**Data Availability Statement:** The data used in the analysis is accessible at: https://clingen.igib.res.in/dalia/DALIA_Table.xlsx.

**Funding:** The work was funded by the Council of Scientific and Industrial Research, India [RareGen

# DALIA- a comprehensive resource of Disease Alleles in Arab population

**Aastha Vatsyayan**[1,2☯‡], **Parul Sharma**[3☯‡], **Shrey Gupta**[3], **Sumiti Sandhu**[4], **Seetha Lakshmi Venu**[5], **Vandana Sharma**[6], **Bouabid Badaoui**[7], **Kaidi Azedine**[7], **Serti Youssef**[7], **Anna Rajab**[8], **Alaaeldin Fayez**[9], **Seema Madinur**[10], **Anop Ranawat**[1], **Kavita Pandhare**[1], **Srinivasan Ramachandran**[1,11], **Sridhar Sivasubbu**[1,11], **Vinod Scaria**[1,11]*

**1** CSIR Institute of Genomics and Integrative Biology (CSIR-IGIB), Delhi, India, **2** Amity University, Noida, India, **3** Indraprastha Institute of Information Technology Delhi (IIIT-D), Delhi, India, **4** Amity University, Rajasthan, Jaipur, India, **5** ACBR, Delhi, India, **6** Kalindi College, University of Delhi, New Delhi, India, **7** Mohammed V University, Rabat, Morocco, **8** National Genetic Center, Ministry of Health, Muscat, Oman, **9** Human Genetics and Genome Research Division, Center of Excellence for Human Genetics, National Research Center, Cairo, Egypt, **10** Independent Researcher, India, **11** Academy of Scientific and Innovative Research (AcSIR), Ghaziabad, India

☯ These authors contributed equally to this work.
‡ These authors share first authorship on this work.
* vinods@igib.in

## Abstract

The Arab population encompasses over 420 million people characterized by genetic admixture and a consequent rich genetic diversity. A number of genetic diseases have been reported for the first time from the population. Additionally a high prevalence of some genetic diseases including autosomal recessive disorders such as hemoglobinopathies and familial mediterranean fever have been found in the population and across the region. There is a paucity of databases cataloguing genetic variants of clinical relevance from the population. The availability of such a catalog could have implications in precise diagnosis, genetic epidemiology and prevention of disease. To fill in the gap, we have compiled DALIA, a comprehensive compendium of genetic variants reported in literature and implicated in genetic diseases reported from the Arab population. The database aims to act as an effective resource for population-scale and sub-population specific variant analyses, enabling a ready reference aiding clinical interpretation of genetic variants, genetic epidemiology, as well as facilitating rapid screening and a quick reference for evaluating evidence on genetic diseases.

## Introduction

The Arab population encompasses twenty three Arab States with a total population of around 420 million individuals and roughly encompassing 5% of the world population [1, 2]. The rich genetic diversity in the Arab population has been contributed by a genetic admixture resulting from waves of migration, to form one of the largest geocultural units in the world [3]. A number of genetic diseases are prevalent in the Arab population. These include

Grant]. The funding body had no role in the preparation of the manuscript or decision to publish. AV acknowledges a Senior Research Fellowship from the Indian Council for Medical Research.

**Competing interests:** The authors have declared that no competing interests exist.

Hemoglobinopathies such as sickle cell disease (HbS) as well as thalassemias, Familial Mediterranean Fever, Osteopetrosis Syndromes, glucose-6-phosphate dehydrogenase (G6PD) deficiency and several metabolic disorders [4]. The prevalence of genetic disorders in the Arab population has been extensively reviewed by a number of authors [4–8]. It is widely believed that datasets with well-curated and annotated genetic datasets are urgently required to establish disease epidemiology, especially in genetically isolated subpopulations which bear a particularly heavy disease burden [4].

A few efforts in the past have attempted to systematically catalog the genetic diseases in the Arab population. The first and foremost major initiative was the Arab Genetic Disease Database, put together by Teebi and colleagues [9]. This was followed by a more systematic effort in later years, resulting in the creation of the Catalogue of Transmission Genetics in Arabs (CTGA), a comprehensive resource mapping genetic diseases in Arab population. The database created by the Centre for Arab Genomic Studies (CAGS) [10] provides information about phenotypes and related genes. In addition, a few resources have emerged that have collected the genetic variants in Arab populations. These include the Greater Middle East (GME) initiative [10, 11], which sequenced multiple individuals in the region to create a comprehensive variome resource. In addition, the Qatar genome programme (About Qatar Genome. Retrieved from https://qatargenome.org.qa/node/5) and a dataset of over 1005 Qatari exomes and genomes [12] aim to provide an overview of genetic diversity in the region. More recently, a genome-scale database, al mena [13] was created by our group, integrating datasets to provide a comprehensive allele frequency resource for the population, apart from providing allele frequencies of variants associated with diseases. A few more population-scale initiatives are underway which include the Saudi Human Genome Project [14] and the 1000 Arab Genome Project [14, 15].

With the increasing adoption of sequencing in clinical settings, a number of new genetic variants and diseases are being reported from the region. Nevertheless there has been no structured effort to curate these genetic variants in a systematic format to allow comparison and enable genetic epidemiology analyses. Our previous studies demonstrate how a systematic, manually curated resource of genetic variants could enable the establishment of genetic epidemiology of disease causing variants in the population, with implications in better diagnosis of the disease [13, 14, 16].

To fill in the gap, we have created a comprehensive manually curated resource of genetic variants in the Arab population. This database, DALIA—for Disease Alleles in Arabs provides a ready reference to genetic variants published from Arab populations for clinicians, patients, as well as researchers. In addition, the resource also serves as a resource for genetic epidemiology.

## Material and methods

### Literature coverage and curation

A list of relevant publications was retrieved using pubmed.mineR [17] tool using country names to query for publications describing "mutation", "variant" or "polymorphism". An exhaustive list of PubMed IDs was retrieved for each of the 23 countries which have a significant number of Arabs. These include 22 countries which were part of the Arab League and speak Arab- Algeria, Bahrain, the Comoros Islands, Djibouti, Egypt, Iraq, Jordan, Kuwait, Lebanon, Libya, Morocco, Mauritania, Oman, Palestine, Qatar, Saudi Arabia, Somalia, Sudan, Syria, Tunisia, the United Arab Emirates, Yemen apart from Israel which has a significant number of Arabs [18–23].

Each of the full-text articles were retrieved and manually curated, to include an extensive array of information, including country of origin and ethnicity apart from the variant type, the methods used for the annotation of variants etc. Special emphasis was made on including only papers that described variants in the countries, and not just reported them.

Each of the variants were further normalised to the GRCh37/hg19 build version of the Human genome for the genomic location and variant position, as well as according to the Human Genome Variation Society (HGVS) nomenclature for the representation of the variants. Gene names were similarly normalised to the Human Gene Nomenclature Committee (HGNC) nomenclature using Mutalyzer tool [24] which check for consistency for the normalised variants. Wherever applicable, dbSNP IDs and ClinVar IDs were added for variants mapping to respective databases. The disease names were also normalised according to the annotations in the Online Mendelian Inheritance in Man (OMIM) [25–27] database and linked wherever they could be consistently mapped.

For each of the variants, additional information on the zygosity of variant, and the method or assay used for identification of the variant were also collected. To ease the annotation, a prefilled list of techniques was used as a drop-down. The entire activity was performed on a template available to all annotators through Google docs online as well as offline through templates in Microsoft Excel. The annotators individually filled in variants for each country. Along with the template, the annotators were also given tutorial slides to train themselves on the curation guidelines so as to maintain uniformity across all curators.

## Quality control and compilation of annotations

Each of the annotation sheets corresponding to the variants from each annotating group were compiled. Each of these sets were then independently cross-checked by a different team member to eliminate manual errors. The corrected sets were compiled into the master sheet. Each of the entries were verified for (i) accuracy and consistency of annotations as per the HGVS nomenclature for variants. (ii) accuracy of the variant position and reference and alternate genotypes on the hg19 genome build and (iii) uniformity and consistency of annotations, using the Mutalyzer tool.

## Allele frequencies and genetic epidemiology

The allele and genotype frequencies of the variants were systematically compiled for the global as well as Middle Eastern populations. The choice of Middle Eastern datasets was based on the fact that no specific datasets are publicly available representative of the Arab populations and the dataset from the region could at least in part provide insights into the prevalence of variants in the region. The global allele frequencies were derived from the ExAC [28] as well as 1000 Genomes [29] databases, while the allele frequencies in Middle Eastern populations were compiled from the al mena database. The al mena database encompasses data from the Qatar subpopulations representing African, Arabian, Bedouin, Persian and South Asian ancestry subgroups as well as the data from the subpopulations which were represented in the Greater Middle East (GME) study [11] representing Asian Peninsula, Central Asian, Israel, Northeast Africa, Northwest Africa, Syrian Desert and Turkish Peninsula. The comparisons were tested for significance using Fisher's Exact Test. Comparisons were done for each subpopulation vis-a-vis the population average and with the global frequencies as derived from ExAC and 1000 Genomes databases.

## Analysis of genes under natural selection

We used two different metrics, the Integrated Haplotype Score (iHS) and Fixation index statistic (Fst) to identify genes under natural selection. The top 1% genes sorted by the iHS scores in

Qatar population were searched for the genes in the database. Pairwise Fst scores [30] were then computed for pairs of Qatari population (QALL) and the African (QAFR), Arab (QARB), Bedouin (QBED), Persian (QPER), and South Asian (QSAS) subpopulations.

### Database and web interface design

The database was developed using MongoDB, a popular open-source NOSQL database system in view of the flexibility offered and the scalability for population-scale variant curation. The web server was configured in Apache 2.4.12, and the interface was coded in AngularJS and PHP.

## Results & discussion

### Data compilation

We compiled a total of 3577 genetic variants from 368 genes and associated with 1984 diseases from over 1113 publications originating from 23 countries. Of these variants, a total of 2790 variants reported were unique, out of which a total of 2679 variants (96%) were exonic, while the remaining included 110 non-exonic (26 3'UTR, 24 intronic, 17 splicing, 12 UTR5, 11 upstream, 1 ncRNA intronic, and 1 a downstream variants).

Apart from the genetic context, associated information on the gene, disease, population from which the variant was reported, and a spectrum of computational scores predicting pathogenicity of the variants, including SIFT [31] Polyphen [27] and CADD [25] were also compiled for each of the variants. Allele frequencies of the variants across population-scale datasets like gnomAD [32] along with linkouts to other relevant databases including OMIM, ClinVar [33] and dbSNP [34] were also compiled systematically.

### Database features

The database is designed to have a user-friendly interface to the data compilation. The prospective user can search the resource in multiple query formats, including genomic location, rsID or gene name. Population-specific variants can also be viewed by searching the name of the country. A complete list of example formats is available on the homepage (Fig 1). A comprehensive result containing details regarding the gene, disease caused, ethnic and geographic details of the patient, along with various allele frequencies as well as variant annotation scores have been linked to each search query (Fig 2).

### Comparison with other databases

The variants in DALIA were compared with variants in ClinVar and the HGMD-Public version to see whether the variants covered in the present compilation are also similarly covered in other databases. Comparison of the 2790 unique genetic variants in dalia with Clinvar revealed that 2074 variants were shared by ClinVar. A similar comparison of variants with HGMD revealed 2197 variants were shared with the Public version of HGMD.

### Allele frequency analysis

We performed a comparison of the allele frequencies of the variants across the population scale datasets of allele frequencies from 1000 Genomes, ExAC, Greater Middle Eastern, and Qatar. The former two represented global populations while the latter two represented population data from the middle east. The Qatar and GME allele frequencies were retrieved from the al mena database. Of the variants, a total of 564 variants had a reported allele frequency in at least one of the four population scale datasets considered, while 255 had frequencies across all the four datasets

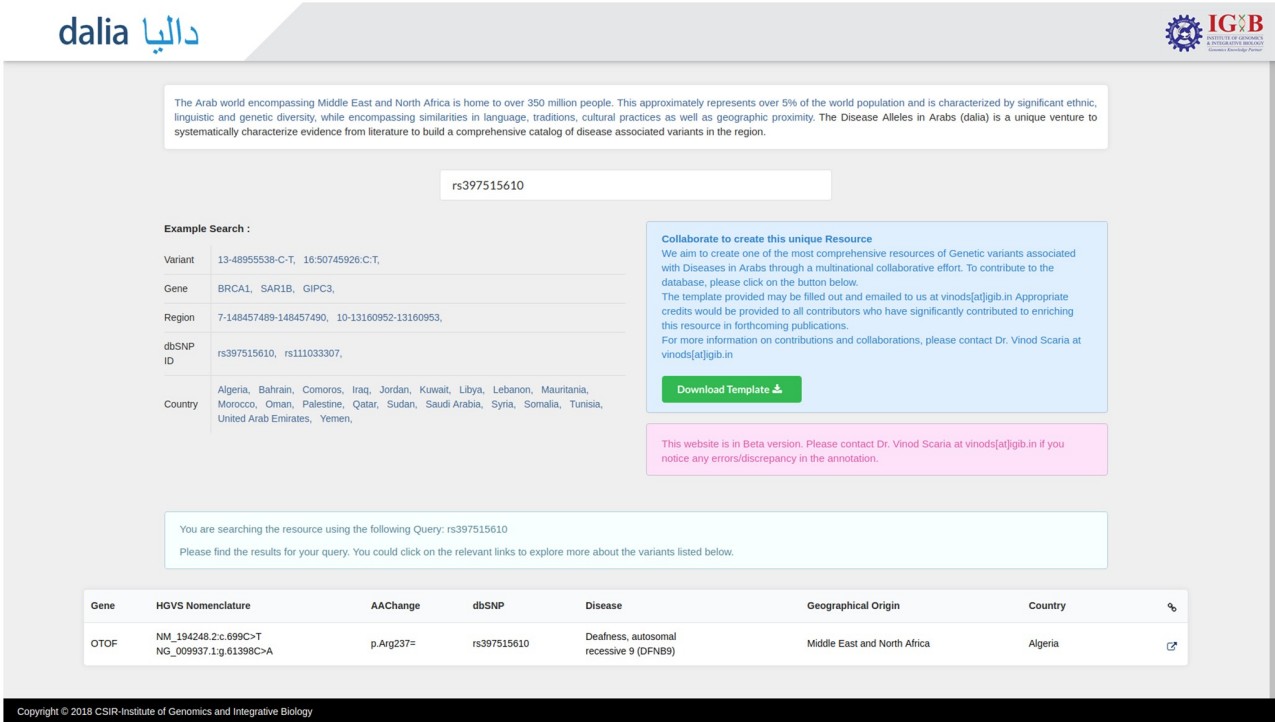

**Fig 1. The DALIA homepage.** It provides an easy to use interface for searching and retrieving the compiled information on genetic variants. The 'Example Search' section illustrates numerous query formats in which the database can be explored.

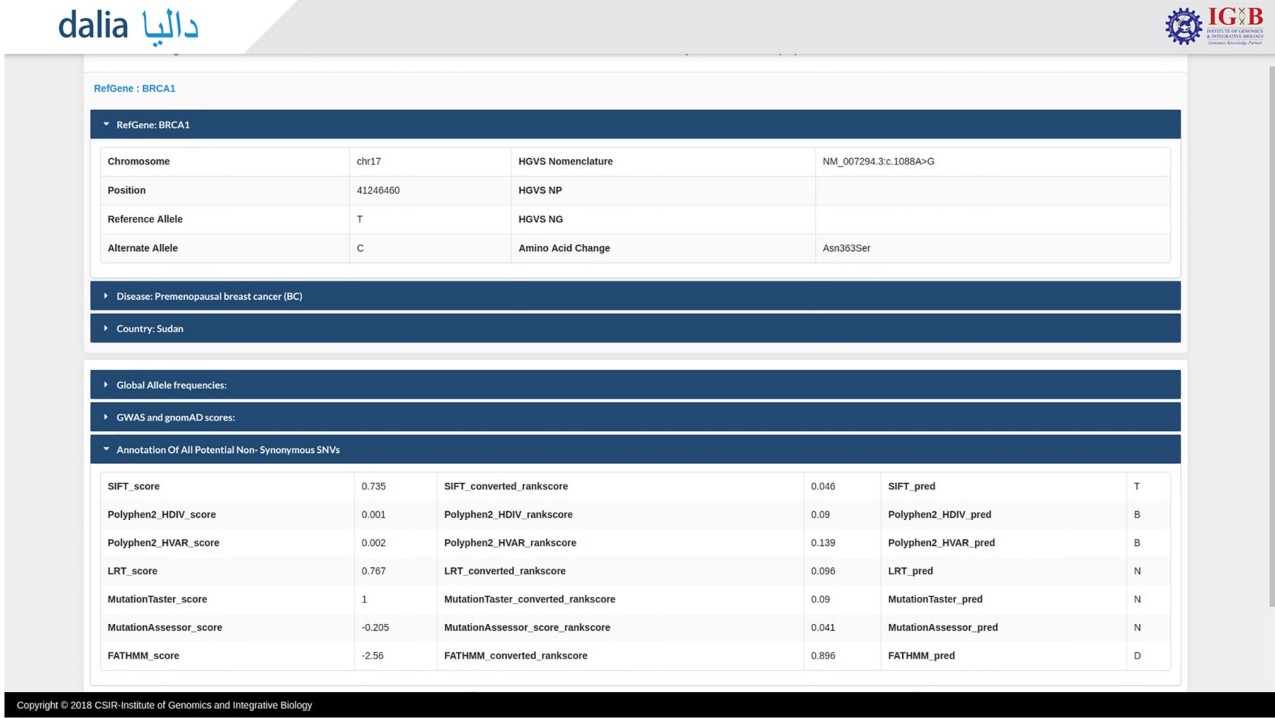

**Fig 2. Expanded search results.** Each selected search result links to a page containing complete information about the variant, including information about the gene, disease, genome level information as well as allele frequencies and various predictive scores that could help establish pathogenicity. Further, links to various other resources including Pubmed, Clinvar, dbSNP and OMIM are also included for a more complete picture of the variant.

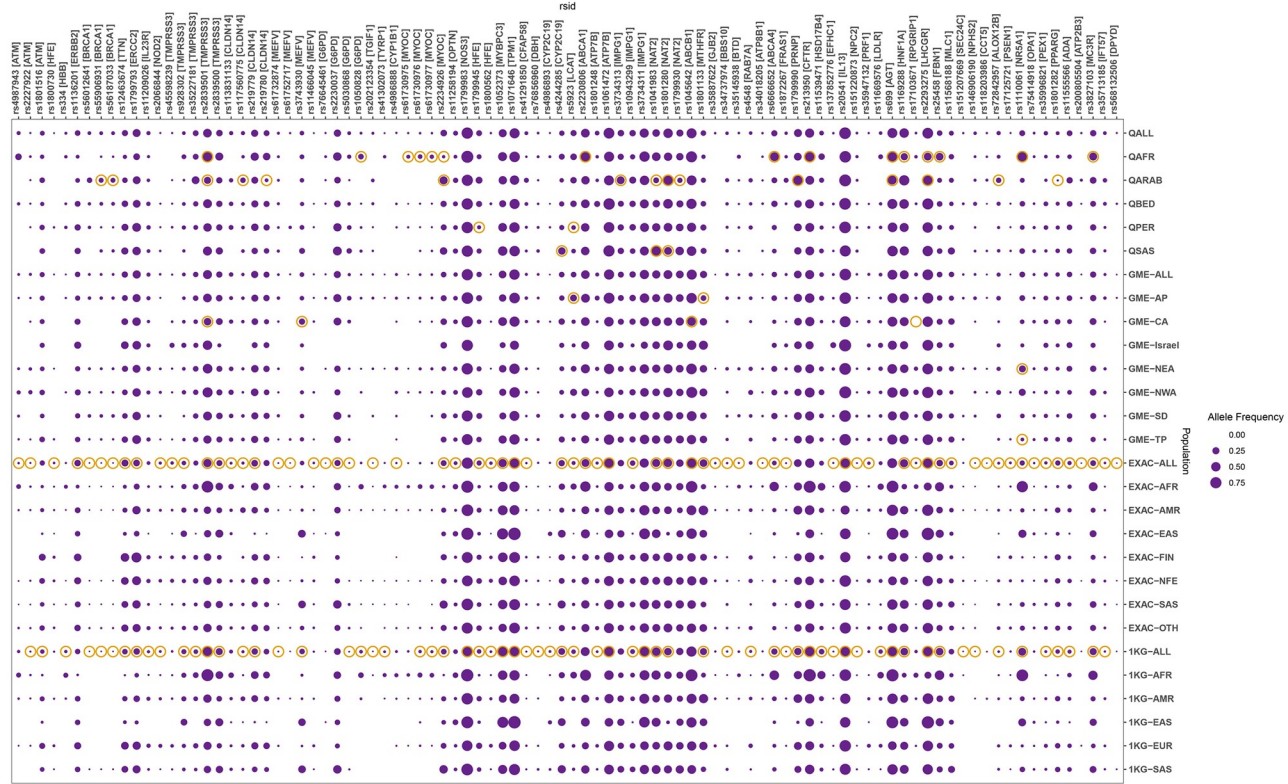

**Fig 3. Comparison of allele frequencies.** The figure represents a subset (p-value < 0.01) of all statistically significant variants obtained after applying the Fisher's Exact Test (S1 Fig). The orange circles highlight variants that were significantly different either within the Qatar or GME subpopulations, or with respect to global population averages ExAC and 1000 Genomes. The populations shown are, in order: Qatari population (QALL) and sub-populations African (AFR), Arab (QARB), Bedouin (QBED), Persian (QPER), and South Asian (QSAS), GME population (GME-ALL), and sub-populations northwest Africa (GME-NWA), northeast Africa (GME-NEA), Arabian peninsula (GME-AP), Israel (GME-Israel), Syrian desert (GME-SD), Turkish peninsula (GME-TP) and Central Asia (GME-CA), ExAC population (EXAC-ALL), sub-populations African/African American (EXAC-AFR), Latino (EXAC-AMR), East Asian (EXAC-EAS), Finnish (EXAC-FIN), Non-Finnish European (EXAC-NFE), South Asian (EXAC-SAS) and Other (EXAC-OTH), 1000 Genomes population(1KG-ALL) and sub-populations African(1KG-AFR), Ad Mixed American(1KG-AMR), East Asian(1KG-EAS), European(1KG-EUR) and South Asian(1KG-SAS).

considered. Statistical analyses were performed for the allele frequencies (S1 Fig) and analysis revealed a total of 142 out of the 255 variants considered had a significant difference with global populations (p-value < 0.01). The variants and their frequencies are summarised in Fig 3.

## Genes and variants under natural selection

Since the individual-level genotypes were available only for the Qatar population, we analysed whether any of the genes harbouring the genetic variants showed signals of natural selection. Briefly, after retrieving the genes in the top 1% of |iHS| scores which encompass 1368 genes, we obtained a total of 10 genes which overlapped with the genes in our compendium. Pairwise Fst scores were further computed for each of the variants in the genes. The 10 genes in the DALIA database had a total of 12 genetic variants reported from the Arab population. The genes and annotations are summarised in S1 Table. Of specific mention would be the ATP7B gene which is associated with Wilson's disease. A total of 37 variants in the ATP7B gene associated with Wilson disease were reported in the DALIA database. The variants had an allele frequency ranging from 0 across QAFR, QARAB, QPER AND QSAS to 0.777 in the QBED Qatar sub-population, and from 0 across GME-CA, GME-NWA and GME-TP subpopulations, to

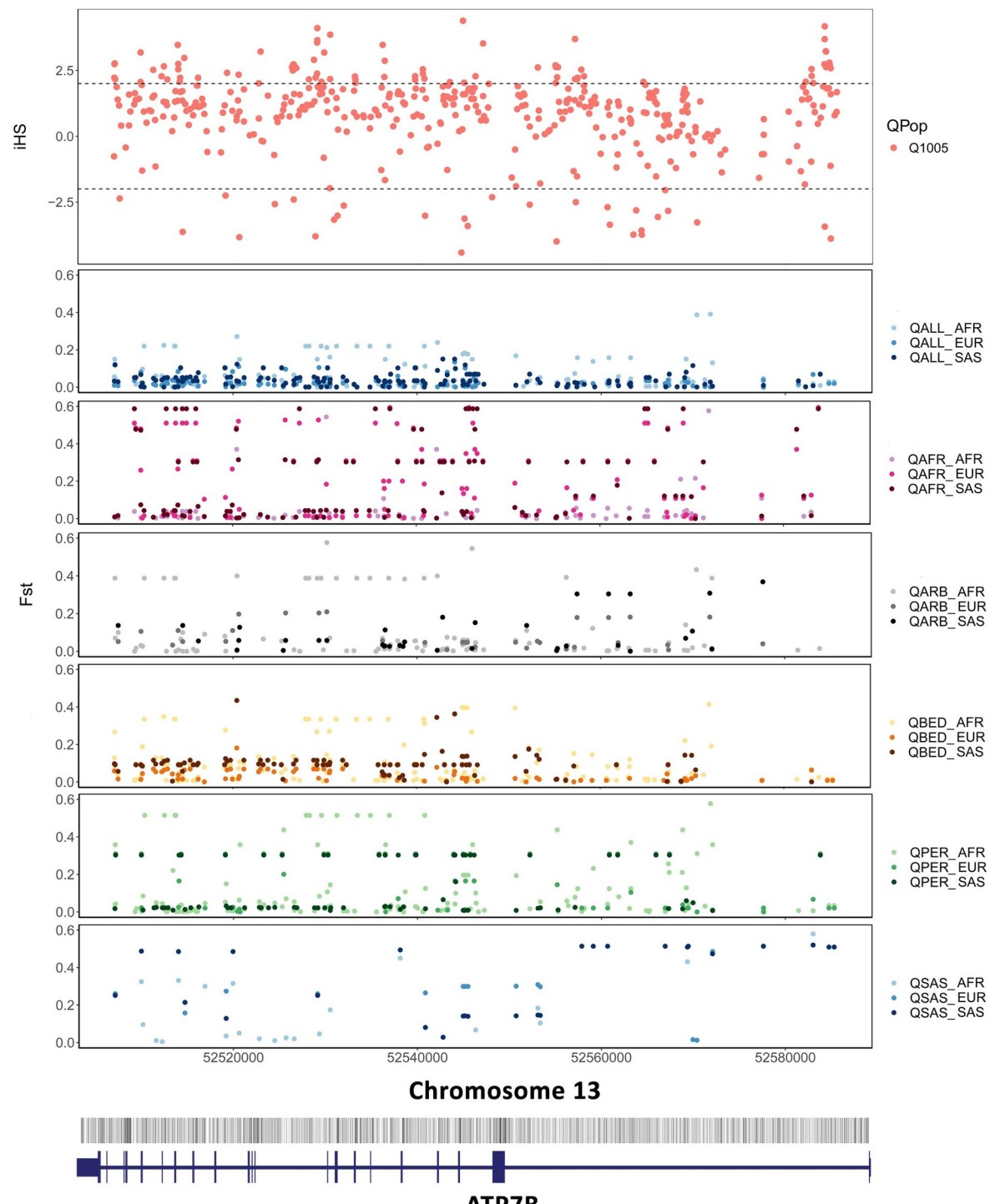

**Fig 4. Signatures of positive selection in the ATP7B gene in the Qatar population.** The figure depicts the iHS scores (top) for all known variants of the ATP7B gene plotted along the length of the gene loci shown below. The subsequent boxes depict all the pairwise Fst scores for the Qatar population as a whole (QALL), as well as all its subpopulations (QAFR, QARAB, QBED, QPER, and QSAS). The array of lines at the bottom represents all known variants along the length of the gene, the exon structure of which is shown beneath it.

**Table 1. G6PD variants in the DALIA database.**

| Variant | HGVS nomenclature | ACMG Classification | Evidence | ExAC | Q1005 | GME | 1000 Genomes |
|---------|-------------------|---------------------|----------|------|-------|-----|--------------|
| rs5030868 | NM_001042351.1: c.563C>T | Pathogenic | PP3, BS2, PP5, PS3, PP1-M, PM1 | EXAC-ALL: 0.003 EXAC-AMR: 0.001 EXAC-EAS: 0 EXAC-FIN: 0 EXAC-NFE:0.001 EXAC-OTH: 0.011 EXAC-SAS: 0.016 | QALL: 0.042 QAFR: 0.061 QARAB: 0.004 QBED: 0.032 QPER: 0.079 QSAS: 0.122 | GME-ALL: 0.024 GME-AP: 0.077 GME-CA:0.042 GME-Israel:0 GME-NEA: 0.008 GME-NWA:0 GME-SD:0.018 GME-TP:0.008 | 1KG-ALL: 0.001 1KG-AFR: 0 1KG-AMR: 0 1KG-EAS: 0 1KG-EUR: 0 1KG-SAS: 0.004 |
| rs76645461 | NM_000402.3: c.233T>C | Likely pathogenic | PP3, PP5, PS3, PM1 | EXAC-ALL: 0 EXAC-AMR: 0 EXAC-EAS: 0 EXAC-FIN: 0 EXAC-NFE: 0 EXAC-OTH: 0 EXAC-SAS: 0 | QALL: 0.004 QAFR: 0.014 QARAB: 0.004 QBED: 0.004 QPER: 0 QSAS: 0 | GME-ALL: 0.003 GME-AP: 0.012 GME-CA: 0 GME-Israel: 0 GME-NEA: 0.003 GME-NWA: 0 GME-SD: 0 GME-TP: 0 | 1KG-ALL: 0 1KG-AFR: 0 1KG-AMR: 0 1KG-EAS: 0.001 1KG-EUR: 0 1KG-SAS: 0 |
| rs5030869 | NM_001360016.2: c.1003G>T | Likely pathogenic | PP3, PP5, PS3, PM1 | EXAC_ALL: 0.0002 EXAC_AFR: 0 EXAC_AMR: 0.0001 EXAC_EAS: 0.0001 EXAC_FIN: 0 EXAC_NFE: 0.0001 EXAC_OTH: 0.001 EXAC_SAS: 0.0006 | QALL: 0.00 QAFR: 0.00 QARAB: 0.00 QBED: 0.00 QPER: 0.01 QSAS: 0.00 | GME_ALL: 0.00 GME_NWA: 0.00 GME_NEA: 0.00 GME_AP: 0.01 GME_Israel: 0.00 GME_SD: 0.00 GME_TP: 0.00 GME_CA: 0.01 | |
| rs2515904 | NM_000402.3:c.576-60C>G | Benign | PP5, BS1, BS2 | | QALL: 0.03 QAFR: 0.19 QARAB: 0.00 QBED: 0.00 QPER: 0.00 QSAS: 0.00 | | 1KG-ALL: 0.05 1KG-AFR: 0.17 1KG-AMR: 0.01 1KG-EAS: 0.00 1KG-EUR: 0.00 1KG-SAS: 0.00 |
| rs5030872 | NM_001360016.2: c.542A>T | Likely pathogenic | BP4, PP5, PS3, PM1 | EXAC_ALL: 0.00 EXAC_AFR: 0.00 EXAC_AMR: 0.00 EXAC_EAS: 0.00 EXAC_FIN: 0.00 EXAC_NFE: 0.00 EXAC_OTH: 0.00 EXAC_SAS: 0.00 | | GME_ALL: 0.00 GME_NWA: 0.01 GME_NEA: 0.00 GME_AP: 0.00 GME_Israel: 0.00 GME_SD: 0.00 GME_TP: 0.00 GME_CA: 0.00 | |

*(Continued)*

**Table 1.** (Continued)

| Variant | HGVS nomenclature | ACMG Classification | Evidence | ExAC | Q1005 | GME | 1000 Genomes |
|---|---|---|---|---|---|---|---|
| rs137852339 | NM_001360016.2: c.949G>A | Likely pathogenic | BP4, PP5, PS3, PP1-M, PM1 | EXAC_ALL: 0.00 EXAC_AFR: 0.00 EXAC_AMR: 0.00 EXAC_EAS: 0.00 EXAC_FIN: 0.00 EXAC_NFE: 0.00 EXAC_OTH: 0.00 EXAC_SAS: 0.01 | QALL:0.01 QAFR:0.00 QARAB:0.00 QBED:0.01 QPER:0.00 QSAS:0.00 | GME_ALL: 0.00 GME_NWA: 0.00 GME_NEA: 0.00 GME_AP: 0.00 GME_Israel: 0.00 GME_SD: 0.00 GME_TP: 0.00 GME_CA: 0.00 | 1KG-ALL: 0.00 1KG-AFR: 0.00 1KG-AMR: 0.00 1KG-EAS: 0.00 1KG-EUR: 0.00 1KG-SAS: 0.01 |

The table contains all population and subpopulation allele frequencies associated with the four G6PD variants of statistical significance (p-value <0.01) present in the DALIA database as well as the pathogenic variants from other databases included in the heatmap. **Note**: For GME populations, the allele frequency has been calculated only for women.

0.702 in GME-AP sub-population in the GME dataset. The iHS scores and the pairwise Fst scores for the variants are plotted in Fig 4. The iHS and Fst scores for the remaining 9 genes are included in the Supplementary file (S2–S10 Figs).

## Integration of datasets for understanding genetic epidemiology

We further explored whether the dataset could provide insights into the genetic epidemiology of mendelian diseases and traits. To this end, we analysed the allele frequencies of the variants in the Qatar exome/genome dataset as well as the GME datasets. Analysis of genetic variants with significant allele frequency differences revealed a total of 94 variants that were significantly different (Fisher's exact, p-value <0.01) in one or more populations or subpopulations. Upon analysis of these 94 variants, we discovered 4 variants mapping to the G6PD gene, associated with G6PD deficiency or favism, a prevalent disease in the Middle east. To provide a common standard for comparison of the pathogenicity annotations for the variants, we reclassified them according to the ACMG/AMP guidelines for interpretation of sequence variants [35]. Two of the four variants were found to have evidence to qualify being annotated as pathogenic or likely pathogenic. The variants, annotations and allele frequencies in each of the population and sub-populations are summarised in Table 1.

For further analysis, we additionally compiled all known pathogenic or likely pathogenic variants in the G6PD gene reported in Clinvar as well as other public databases. This resulted in a final list of 44 unique variants. We further analysed allele frequencies for these variants from across the four databases. Of the 44 variants, 21 had known allele frequencies in one or more of the four population datasets and 6 had allele frequencies in either Qatar, GME or both of the middle eastern population datasets.

Further analysis of these 6 variants revealed that the variants had allele frequencies ranging from 0.0 to 0.187 across different sub populations. Some of the variants showed significant population specificity, as in the case of rs5030872, which had allele frequencies in GME_ALL (0.001) and GME_NWA (0.01) but has not been reported in any of the Qatar or 1000 genome datasets. Similarly, rs2515904 has allele frequencies reported in Qatar (0–0.187) and 1000genomes (0–0.173) population datasets, but not in the GME and ExAC populations, indicating differences even among the Qatar and GME populations. Some variants showed intrapopulation variation, for example rs2515904, which had allele frequencies in QALL

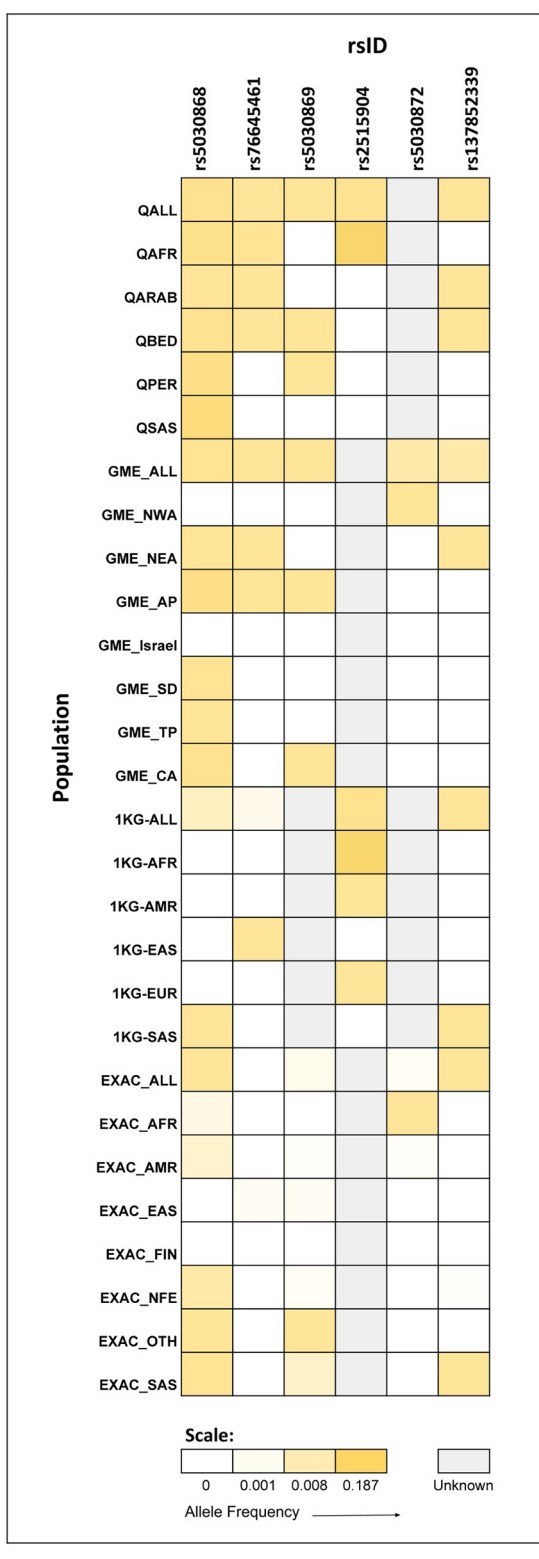

**Fig 5. Allele frequency heat map.** The figure depicts the variation in allele frequency distribution of all known G6PD pathogenic/likely pathogenic variants across four major datasets—Qatar, GME, 1000 Genomes and ExAC.

(0.034) and QAFR (0.187), but had allele frequency 0 in QARAB, QBED, PER and QSAS. These variations are summarized in Fig 5 as an allele frequency heatmap.

## Conclusions

In summary, DALIA fills the need for a relevant, manually curated and annotated database of variants associated with genetic diseases in the Arab populations. The DALIA database provides searchable access to over 2700 genetic variants and associated information published from the region. Apart from being a ready reference to clinicians and clinical geneticists, we also suggest using examples how such a database could provide insights into genetic epidemiology of diseases as well as clues on prevalent genetic variants in subpopulations which is pertinent in developing approaches for cost effective diagnosis and screening of genetic diseases.

With population scale genome programmes now underway in the region [14, 15, 36] it is imperative that DALIA would become an important resource enabling appropriate analysis and interpretation of genomic data as well as a central resource for genetic epidemiology of diseases. Similarly with the advent of whole exome and whole genome sequencing becoming prevalent in clinical settings for the diagnosis of genetic diseases [36–39] it is expected that the accelerated discovery of genetic variants in diseases would also benefit the DALIA database in years to come.

In summary, to the best of our knowledge DALIA is one of the largest compilations of disease associated genetic variants from the Arab populations.

## Supporting information

**S1 Fig. Bubble plot displaying all the 143 significantly different variants found in the DALIA database upon comparison with both regional subpopulations as well as global population averages.**
(TIF)

**S2 Fig. Plot depicting the iHS and pairwise Fst scores along the gene loci, for all the known variants for the gene ABCA4.** The 10 genes with top 1% |iHS| scores include ABCA4, THSD1, ATP7B, MEFV, ADA, CLDN14, MLC1, HSD17B4, NAT2, and ABCA1 (S1 Table). This plot depicts the iHS as well as pairwise Fst scores along the gene loci, for all the known variants for the gene ABCA4. The array of lines at the bottom represent all known variants, and the exon structure of the gene is shown beneath it.
(TIF)

**S3 Fig. Plot depicting the iHS and pairwise Fst scores along the gene loci, for all the known variants for the gene THSD1.**
(TIF)

**S4 Fig. Plot depicting the iHS and pairwise Fst scores along the gene loci, for all the known variants for the gene MEFV.**
(TIF)

**S5 Fig. Plot depicting the iHS and pairwise Fst scores along the gene loci, for all the known variants for the gene ADA.**
(TIF)

**S6 Fig. Plot depicting the iHS and pairwise Fst scores along the gene loci, for all the known variants for the gene CLDN14.**
(TIF)

**S7 Fig. Plot depicting the iHS and pairwise Fst scores along the gene loci, for all the known variants for the gene MLC1.**
(TIF)

**S8 Fig. Plot depicting the iHS and pairwise Fst scores along the gene loci, for all the known variants for the gene HSD17B4.**
(TIF)

**S9 Fig. Plot depicting the iHS and pairwise Fst scores along the gene loci, for all the known variants for the gene NAT2.**
(TIF)

**S10 Fig. Plot depicting the iHS and pairwise Fst scores along the gene loci, for all the known variants for the gene ABCA1.**
(TIF)

**S1 Table. The table provides details of the ten genes in our database that had top 1% |iHS| scores, indicating positive selection.**
(XLSX)

## Acknowledgments

We acknowledge the research community working actively on genetic diseases for their whole-hearted contribution, suggestions and contributions to the database without which it would have been impossible to be able to put together the resource.

## Author Contributions

**Conceptualization:** Sridhar Sivasubbu, Vinod Scaria.

**Data curation:** Aastha Vatsyayan, Shrey Gupta, Sumiti Sandhu, Seetha Lakshmi Venu, Vandana Sharma, Bouabid Badaoui, Kaidi Azedine, Serti Youssef, Anna Rajab, Alaaeldin Fayez, Seema Madinur.

**Formal analysis:** Aastha Vatsyayan.

**Funding acquisition:** Vinod Scaria.

**Investigation:** Aastha Vatsyayan, Vinod Scaria.

**Methodology:** Vinod Scaria.

**Project administration:** Aastha Vatsyayan, Vinod Scaria.

**Resources:** Anop Ranawat, Srinivasan Ramachandran, Vinod Scaria.

**Software:** Anop Ranawat, Kavita Pandhare, Srinivasan Ramachandran, Vinod Scaria.

**Supervision:** Parul Sharma, Vinod Scaria.

**Validation:** Aastha Vatsyayan.

**Visualization:** Aastha Vatsyayan.

**Writing – original draft:** Aastha Vatsyayan.

**Writing – review & editing:** Aastha Vatsyayan, Vinod Scaria.

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
