## [Decision Letter · Decision Letter 0]

8 Oct 2020

PONE-D-20-26943

DALIA- a Comprehensive Resource of Disease Alleles in Arab Population

PLOS ONE

Dear Dr. Scaria,

Thank you for submitting your manuscript to PLOS ONE. After careful consideration, we feel that it has merit but does not fully meet PLOS ONE’s publication criteria as it currently stands. Therefore, we invite you to submit a revised version of the manuscript that addresses the points raised during the review process.

One of the reviewers raised concerns on the way the population is described and the rationale for this study is not clear. Please make those points clear and also the other points raised by the reviewers. Also explain your reasons for the interpretation that large family size results in genetic disorders. Not clear why that would be the case, and what is the evidence that family sizes are large among Arabs providing supporting references also for the source of information on the population size.

We look forward to receiving your revised manuscript.

Kind regards,

Obul Reddy Bandapalli, MSc, PhD

Academic Editor

PLOS ONE

Journal Requirements:

Reviewers' comments:

Reviewer's Responses to Questions

**Comments to the Author**

1. Is the manuscript technically sound, and do the data support the conclusions?

Reviewer #1: No

Reviewer #2: Yes

2. Has the statistical analysis been performed appropriately and rigorously? 

Reviewer #1: I Don't Know

Reviewer #2: Yes

3. Have the authors made all data underlying the findings in their manuscript fully available?

Reviewer #1: Yes

Reviewer #2: Yes

4. Is the manuscript presented in an intelligible fashion and written in standard English?

Reviewer #1: No

Reviewer #2: Yes

5. Review Comments to the Author

Reviewer #1: This paper seeks to describe how a comprehensive database for genetic information on Arab populations was formed, and it's characteristics. The rationale for data base need is strong: heavy burden of genetic disorders in the region underscores the need for genome data analyses. However, the purpose of the paper is not clear. The authors need to state with more clarity what the paper aims to do, and why that aim is important for advancing the field. Other details are elaborated below.

1. There is no source of information in terms of the basic statement that there are twenty-three Arab States with a total population of around 420 million individuals and roughly encompassing 5% of the world population. What are the 23 countries? How is Arab defined?

2. The authors seem to conflate Arab with Middle Eastern. Need to distinguish Arab from Middle Eastern.

3. The authors state that large family size results in genetic disorders. Not clear why that would be the case, and what is the evidence that family sizes are large among Arabs?

4. Supporting information is difficult to make sense of. What does it support and how does that information advance the aims of the paper?

Reviewer #2: DALIA- a Comprehensive Resource of Disease Alleles in Arab Population by Aastha Vatsyayan et al has compiled and made the resource important for genetic analysis of Arab population.

It is quite an impressive work done. The features in the database have been incorporated thoughtfully.

6. PLOS authors have the option to publish the peer review history of their article (what does this mean?). If published, this will include your full peer review and any attached files.

Reviewer #1: No

Reviewer #2: No

---

## [Author Response · Author response to Decision Letter 0]

25 Nov 2020

Response to Reviewers

RESPONSE TO REVIEWER 1

Reviewer #1: This paper seeks to describe how a comprehensive database for genetic information on Arab populations was formed, and it's characteristics. The rationale for data base need is strong: heavy burden of genetic disorders in the region underscores the need for genome data analyses. However, the purpose of the paper is not clear. The authors need to state with more clarity what the paper aims to do, and why that aim is important for advancing the field. Other details are elaborated below.

ANS: We thank the reviewers for the prompt review and suggestions. 

The aim of creating the data collection is to provide a ready reference to genetic variants reported from Arab populations. This is relevant as a significant number of genetic variants with clinical implications are not readily curated and available in global databases like ClinVar, which necessitates the creation of focussed and local databases relevant to the populations. The database as its name suggests, encompasses genetic variants reported from the 23 Arab countries. 

1. There is no source of information in terms of the basic statement that there are twenty-three Arab States with a total population of around 420 million individuals and roughly encompassing 5% of the world population. What are the 23 countries? How is Arab defined?

ANS: We thank the reviewer for asking for the clarifications. The 23 Arab countries are defined as per the Arab League [1]. We have now updated the manuscript with more details as well as relevant citations. 

The Arab League is an alliance of Arab countries that aims to foster cooperation in economics, communication, culture, nationality, social welfare, and health and military affairs among its 22 members, namely: Algeria, Bahrain, the Comoros Islands, Djibouti, Egypt, Iraq, Jordan, Kuwait, Lebanon, Libya, Morocco, Mauritania, Oman, Palestine, Qatar, Saudi Arabia, Somalia, Sudan, Syria, Tunisia, the United Arab Emirates, and Yemen [1–4]. Israel, though not a part of this alliance, has a portion of population that comprises Arabs [5,6], however the exact reported total figures vary. These 23 countries are represented in the DALIA database. Though estimates vary, the Arab League has a population of about 427.87 million people, and is home to about 5.6% of the world population [7,8]. All 22 countries form a part of the Arabic speaking world, as does the Arabic population in Israel.

2. The authors seem to conflate Arab with Middle Eastern. Need to distinguish Arab from Middle Eastern.

ANS: We apologise for any confusion regarding this. The data encompasses genetic variants reported in Arab populations. The Allele frequency estimates use the Middle Eastern datasets, as no population scale data for the Arab populations exist in public domain. We have now clarified this in the revised manuscript. 

3. The authors state that large family size results in genetic disorders. Not clear why that would be the case, and what is the evidence that family sizes are large among Arabs?

ANS: We apologise for the confusion. We have now removed the statement in the revised manuscript and added references wherever appropriate to clarify the statements.

4. Supporting information is difficult to make sense of. What does it support and how does that information advance the aims of the paper?

ANS: While the manuscript provides a detailed overview of the DALIA database, The supporting information provides the detailed metrics for the Allele frequencies as well as FST calculations, for the interest of clinicians and genetic epidemiologists with a ready dataset and reference. 

An example of such use would be the analysis of allele frequencies as shown in our paper, in which we compared the variant allele frequencies in Qatar and GME with those in global populations reported in 1000 Genomes and ExAC databases. We obtained 142 statistically significant variants (Fisher’s Exact Test, p-value < 0.01) as a result of this analysis (S1 Fig), 94 of which (Fig 3) were selected for ease of study after applying Bonferroni correction. All significant variants represent potential targets that could be pathogenic for specific populations, and thus make ideal targets for screening in those populations, and consequently for genetic counselling. As evidence, we performed the ACMG annotation of 4 G6PD variants, two of which were found to have evidence to qualify being annotated as pathogenic or likely pathogenic. 

Further, iHS and Fst are two widely used tests to determine positive selection, which can help map changing allele frequencies at population scale. In our work, we found 10 genes that had extreme iHS scores, indicating selection. Each of then 10 plots (Fig 4, S2-S10 Figs) also depicts all known variants along the length of the gene, along with its exon structure. These genes could be linked with diseases of interest - e.g., MEFV is linked to Familial Mediterranean Fever, ABCA4 to Stargardt macular dystrophy [9] and CLDN deafness (OMIM 605608), and would help researchers visualize whether a variant in a positively selected gene is selected, and also lies in the coding region, and what its scores are. 

RESPONSE TO REVIEWER 2

Reviewer #2: DALIA- a Comprehensive Resource of Disease Alleles in Arab Population by Aastha Vatsyayan et al has compiled and made the resource important for genetic analysis of Arab population.

It is quite an impressive work done. The features in the database have been incorporated thoughtfully.

ANS: We thank the reviewer. 

References

1. The state of health in the Arab world, 1990–2010: an analysis of the burden of diseases, injuries, and risk factors. Lancet. 2014;383: 309–320.

2. The Arab League. [cited 21 Nov 2020]. Available: https://www.cfr.org/backgrounder/arab-league

3. Geography of the Modern Middle East and North Africa. [cited 21 Nov 2020]. Available: http://www.middleeastpdx.org/resources/original/geography-of-the-modern-middle-east-and-north-africa/

4. Khachfe HH, Refaat MM. Bibliometric analysis of Cardiovascular Disease Research Activity in the Arab World. International Cardiovascular Forum Journal. 2019. doi:10.17987/icfj.v15i0.554

5. Danial-Farran N, Brownstein Z, Gulsuner S, Tammer L, Khayat M, Aleme O, et al. Genetics of hearing loss in the Arab population of Northern Israel. Eur J Hum Genet. 2018;26: 1840–1847.

6. Brunstein Klomek A, Nakash O, Goldberger N, Haklai Z, Geraisy N, Yatzkar U, et al. Completed suicide and suicide attempts in the Arab population in Israel. Soc Psychiatry Psychiatr Epidemiol. 2016;51: 869–876.

7. Arab Countries 2020. [cited 21 Oct 2020]. Available: https://worldpopulationreview.com/country-rankings/arab-countries

8. Member states of the Arab League. [cited 21 Nov 2020]. Available: https://www.worlddata.info/alliances/arab-league.php

9. Piccardi M, Fadda A, Martelli F, Marangoni D, Magli A, Minnella AM, et al. Antioxidant Saffron and Central Retinal Function in ABCA4-Related Stargardt Macular Dystrophy. Nutrients. 2019;11. doi:10.3390/nu11102461

---

## [Decision Letter · Decision Letter 1]

14 Dec 2020

DALIA- A Comprehensive Resource of Disease Alleles in Arab Population

PONE-D-20-26943R1

Dear Dr. Scaria,

We’re pleased to inform you that your manuscript has been judged scientifically suitable for publication and will be formally accepted for publication once it meets all outstanding technical requirements.

Kind regards,

Obul Reddy Bandapalli, MSc, PhD

Academic Editor

PLOS ONE

Additional Editor Comments (optional):

Reviewers' comments:

Reviewer's Responses to Questions

**Comments to the Author**

1. If the authors have adequately addressed your comments raised in a previous round of review and you feel that this manuscript is now acceptable for publication, you may indicate that here to bypass the “Comments to the Author” section, enter your conflict of interest statement in the “Confidential to Editor” section, and submit your "Accept" recommendation.

Reviewer #2: All comments have been addressed

2. Is the manuscript technically sound, and do the data support the conclusions?

Reviewer #2: Yes

3. Has the statistical analysis been performed appropriately and rigorously? 

Reviewer #2: Yes

4. Have the authors made all data underlying the findings in their manuscript fully available?

Reviewer #2: Yes

5. Is the manuscript presented in an intelligible fashion and written in standard English?

Reviewer #2: Yes

6. Review Comments to the Author

Reviewer #2: The authors have answered the reviewers concern. No additional comments. Also the data related to database has been included

7. PLOS authors have the option to publish the peer review history of their article (what does this mean?). If published, this will include your full peer review and any attached files.

Reviewer #2: No

---

## [Editor Report · Acceptance letter]

4 Jan 2021

PONE-D-20-26943R1 

DALIA- a Comprehensive Resource of Disease Alleles in Arab Population 

Dear Dr. Scaria:

I'm pleased to inform you that your manuscript has been deemed suitable for publication in PLOS ONE. Congratulations! Your manuscript is now with our production department. 

Kind regards, 

on behalf of

Dr. Obul Reddy Bandapalli 

Academic Editor

PLOS ONE